# Structural and mechanistic basis of differentiated inhibitors of the acute pancreatitis target kynurenine-3-monooxygenase

Jonathan P. Hutchinson[1], Paul Rowland[1], Mark R.D. Taylor[2], Erica M. Christodoulou[1], Carl Haslam[1], Clare I. Hobbs[1], Duncan S. Holmes[3], Paul Homes[1], John Liddle[3], Damian J. Mole[4,5], Iain Uings[3], Ann L. Walker[3], Scott P. Webster[6], Christopher G. Mowat[2] & Chun-wa Chung[1]

Kynurenine-3-monooxygenase (KMO) is a key FAD-dependent enzyme of tryptophan metabolism. In animal models, KMO inhibition has shown benefit in neurodegenerative diseases such as Huntington's and Alzheimer's. Most recently it has been identified as a target for acute pancreatitis multiple organ dysfunction syndrome (AP-MODS); a devastating inflammatory condition with a mortality rate in excess of 20%. Here we report and dissect the molecular mechanism of action of three classes of KMO inhibitors with differentiated binding modes and kinetics. Two novel inhibitor classes trap the catalytic flavin in a previously unobserved tilting conformation. This correlates with picomolar affinities, increased residence times and an absence of the peroxide production seen with previous substrate site inhibitors. These structural and mechanistic insights culminated in GSK065(C1) and GSK366(C2), molecules suitable for preclinical evaluation. Moreover, revising the repertoire of flavin dynamics in this enzyme class offers exciting new opportunities for inhibitor design.

[1] Platform Technologies and Science, GlaxoSmithKline, Stevenage SG1 2NY, UK. [2] EastChem School of Chemistry, University of Edinburgh, Edinburgh EH9 3FJ, UK. [3] Discovery Partnerships with Academia, GlaxoSmithKline, Stevenage, UK. [4] Medical Research Council Centre for Inflammation Research, Edinburgh EH16 4TJ, UK. [5] Clinical Surgery, University of Edinburgh, Edinburgh EH16 4SA, UK. [6] Centre for Cardiovascular Science, University of Edinburgh, Edinburgh EH16 4TJ, UK. Correspondence and requests for materials should be addressed to C.-w.C. (email: cc16943@gsk.com).

I n mammals, tryptophan is an essential amino acid and the major metabolic route of tryptophan degradation in peripheral tissues is via the kynurenine pathway[1,2]. A number of the metabolites of this pathway have biological activity and imbalances in their levels have been implicated in neurological disorders such as Alzheimer's[3] and Huntington's[4], as well as inflammatory indications including multiple sclerosis[5] and acute pancreatitis[6,7]. Kynurenine-3-monooxygenase (KMO) is a class A flavoprotein monooxygenase (FPMO) which catalyses the hydroxylation of L-kynurenine (L-Kyn) to 3-hydroxykynurenine (3-HK), using NADPH as a co-substrate. It lies at a key branch point of the kynurenine pathway and its expression is largely restricted to the liver, kidney, some monocytic cells, and to a lesser extent in the microglia within the brain. 3-HK is widely described as a neurotoxin that contributes not only to neurodegenerative disorders, but also oxidative stress, for example, in the eye during cataract formation, and can induce apoptotic cell death in endothelial cells[8].

The enzymatic mechanism of KMO has been most extensively characterized using the *Pseudomonas fluorescens* (Pf) bacterial system (Fig. 1)[9]. In the reductive half of the catalytic cycle, kynurenine and NADPH are bound and flavin reduction occurs by hydride transfer. Dissociation of the first product (NADP$^+$) completes the half-reaction. The rate of flavin reduction is greatly increased by the presence of kynurenine within the complex, (by a factor of $\sim 2.5 \times 10^3$ in *P. fluorescens*), a feature which acts as a gating mechanism to prevent uncoupled NADPH turnover[9]. In the oxidative half reaction, molecular oxygen reacts with the reduced flavin to form a highly reactive, but weakly electrophilic C4a-hydroperoxide intermediate, which hydroxylates the activated aromatic ring of L-Kyn leaving a C4a-hydroxy flavin species. Elimination of water from this species returns the flavin to the oxidized state. 3-hydroxykynurenine dissociation completes the cycle and is believed to be the rate limiting step for Pf-KMO[9].

Pharmacological inhibition of KMO has shown promise in animal models of Alzheimer's and Huntington's disease ameliorating neurodegeneration, preventing spatial memory deficits, anxiety-related behaviour and synaptic loss[2,3], although this appears not to be sufficient to alter progression in chronic disease[10]. Recently, we demonstrated the beneficial role of KMO inhibition in the pathogenesis of acute pancreatitis[6]. This disease is typically caused by gallstones or excessive alcohol consumption. For the majority of acute pancreatitis patients, symptoms resolve within a week or so. However, in approximately one in five cases, the disease progresses to multiple organ failure with a mortality rate that exceeds 20% (refs 7,11). Currently no disease modifying treatments are available and medical care is solely supportive which indicates a high unmet medical need.

The first discovered KMO inhibitors were analogues of the substrate L-Kyn which stimulated the production of cytotoxic hydrogen peroxide by triggering uncoupled NADPH oxidation within the KMO catalytic cycle[9]. Here we describe the molecular mechanism of three novel classes of KMO inhibitors with distinct binding modes and kinetics. Two of these classes trap the flavin in a previously unseen 'tilting' conformation, which correlates with picomolar affinities and increased residence times. We term these type II KMO inhibitors and offer a structural rationale for the absence of peroxide production despite occupation of the substrate site. We believe these are the first reported inhibitors that exploit this flavin motion in class A FPMOs, suggesting targeting flavin dynamics may be a productive opportunity for novel inhibitor design.

## Results

### Pf-KMO as a structural surrogate for inhibitor design. KMO has only 10% sequence identity to the prototypical enzyme in this class, p-hydroxybenzoate hydroxylase (PHBH) and on initiation

of our studies no structure of KMO had been determined. Human KMO is a 56 kDa protein containing a tightly bound FAD cofactor and is attached to the mitochondrial outer membrane via a C-terminal region. While we were able to produce human protein suitable for mechanistic studies[12], attempts to generate a human form suitable for crystallography were unsuccessful; the crystal structure of human KMO is still unknown. We therefore turned our attention to the homologous bacterial *P. fluorescens* enzyme, Pf-KMO, which shares 34% sequence identity with the human enzyme, but lacks the transmembrane region and had been previously extensively characterized by Crozier-Reabe et al.[9] (Supplementary Fig. 1).

Initial substrate bound Pf-KMO crystals showed poor X-ray diffraction. Subsequent growth optimization and the generation of a double C252S/C461S Pf-KMO-mutant enabled us to solve our first structure using a selenomethionine-labelled protein to a modest resolution of 3.4 Å (ref. 6). Further rounds of improvement were successful in delivering a high resolution-diffracting monoclinic crystal form with two holoenzyme molecules in the asymmetric unit that was amenable to ligand soaking studies. This system was adopted to generate the substrate bound Pf-KMO structure shown in Fig. 2 and for subsequent inhibitor complexes.

The crystal structure of Pf-KMO in complex with substrate was determined at 1.5 Å resolution (Fig. 2). The KMO monomer consists of three domains: the $\alpha + \beta$ FAD-binding domain, an $\alpha + \beta$ domain containing a six-stranded, mostly antiparallel, β-sheet and a C-terminal four-helix bundle domain (Fig. 2b). The active site is predominantly lined by the small hydrophobic side chains of the six-stranded β-sheet on one side (domain 2) and on the other by the backbone of the P318-Q322 loop and residues from the long α-helix (domain 1) which leads to the C-terminal domain 3 (Fig. 2a,b). Substrate binds in the active site pocket, at the interface of the first two domains, with the aromatic ring lying adjacent and perpendicular to the flavin isoalloxazine ring, the aminobenzaldehyde ring being closest to the FAD, and the carboxylate chain leads out towards the opening of the active site entrance channel (Fig. 2b,c). The ring occupies a pocket lined mostly by hydrophobic residues (F238, P318, F319, H320, G321, A56, I106, L213 and M373) while the carboxylate and amino substituents interact with hydrophilic residues; the carboxylate forms a salt bridge with R84 and the amino group hydrogen bonds to the sidechain of Y98 (Fig. 2c,d). All the active site residues are conserved between the *P. fluorescens* and human enzymes, with the exception of H320, which is a phenylalanine in the human protein and a tyrosine in yeast (Fig. 2, Supplementary Fig. 1). This structural similarity is mirrored by comparable kynurenine $K_m$ values for *P. fluorescens* and human enzymes (Fig. 2f).

A chloride ion could be modelled into electron density on the *re* face of the flavin cofactor (Fig. 2e). Sitting within a curved stretch of backbone, the halide makes interactions with the amides of G323 and Q322 on one side, the hydrophobic ring of P318 on another and two waters at 3.0 and 3.2 Å complete this coordination. This constellation of interactions bears a striking resemblance to that seen for chloride ions in FAD-dependent tryptophan halogenases such as PrnA[13] and RebH[14] where the halide acts as a co-substrate (Supplementary Fig. 2). In flavin hydroxylases such as 3-hydroxybenzoate 6-hydroxylase[15] and KMO the chloride ion appears to play a more passive role, as a placeholder for the space predicted to be occupied by the oxygen atoms of the flavin-hydroperoxide intermediate. In this context, many flavoprotein hydroxylases, including KMO, are inhibited by chloride ions[15,16].

While the structure of human KMO has remained elusive, that of the yeast orthologue has recently been reported in a holo and inhibitor (UPF648) bound form; no substrate complex could be obtained[17]. Reassuringly the yeast protein has a similar

**Figure 1 | Catalytic cycle of KMO.** Schematic of the sequence of catalytic steps involved in the hydroxylation of L-Kyn to 3-HK by the FAD-dependent KMO enzyme[9]. For clarity, chemical structures are drawn only for the isoalloxazine ring of the catalytic flavin and the L-Kyn substrate.

architecture to that of our bacterial enzyme, but as a two domain yeast truncate was crystallized, this lacks insights into the role of the C-terminal domain 3 present in all three native enzymes and unique amongst FPMOs (Supplementary Figs 1 and 3). The Pf-KMO-L-Kyn complex reveals domain 3 further encloses the substrate channel shielding it from bulk solvent (Supplementary Fig. 3) and importantly completes the active site architecture donating residues such as Y404 that directly interact with L-Kyn (Fig. 2d, Supplementary Fig. 3). Contrary to predictions, including a KMO-substrate model based on the yeast structure, the carbonyl of L-Kyn makes no direct interactions with Pf-KMO and is not a key recognition motif. Many substrate-based inhibitors have H-bond acceptor motifs assumed to mimic this carbonyl; in light of this structure any positive contributions these make may need to be re-interpreted by the presence of additional interactions or conformational considerations. The intact Pf-KMO structure therefore provides a more complete picture of the catalytic site and significantly enhances our ability to drive structure-based drug design for KMO.

**Design of substrate-based inhibitors.** Keen to understand the potential of our Pf-KMO surrogate crystal system to guide structural insights into the mechanism and inhibitor mode of action for the human enzyme, we profiled the inhibitor pharmacology across three distinct series of compounds (Fig. 3a). Encouragingly, while inhibitors generally appear less potent in the bacterial enzyme, the rank order of inhibitory activities was conserved between the two species, suggesting despite the low-sequence conservation outside the active site, Pf-KMO has preserved the pharmacology for our inhibitors of interest (Fig. 3b,

Table 1). Consequently Pf-KMO has been used as a structural substitute for human KMO in our structure-based drug design efforts, but the human enzyme has been used for all other *in vitro* and cellular studies (unless explicitly specified).

A number of KMO inhibitor series have been reported[10,18]. The majority share a common pharmacophore containing both an acidic moiety and a mono or 1,2-dichloro substitution of the core phenyl ring, including our previously reported oxazolidinone molecule, GSK180(A1)[6]. Replacement of one chlorine by methyl gave GSK428(A2), which showed a more attractive overall profile[19,20]. These are all believed to share a common mode of action, binding within the substrate site and this was confirmed for GSK428(A2) both enzymatically and by determining X-ray structures of complexes with Pf-KMO (Fig. 4a–d).

Comparison of the bound structures of L-Kyn and GSK428(A2) (Fig. 4b) show the aromatic rings of L-Kyn and GSK428(A2) lie in the same plane. The ring oxygen of the oxazolidinone overlaps with the carbonyl of L-Kyn and similarly makes no direct interactions, whereas the carbonyl of the oxazolidinone hydrogen bonds to Y404 and via a bridging water to the backbone carbonyl of G321. The pendant acid of GSK428(A2) makes slightly different interactions in Chains A and B. In chain A there are direct interactions with Y98 and N369 (Fig. 4a), whereas in Chain B these are water-bridged as seen for the substrate (Figs 2d and 4b).

Enticed by the opportunity to explore a water filled pocket formed by N54, E195, L226 and T236 off the methyl position of GSK428(A2) (Fig. 4d), compounds were made with a variety of substitutions to probe the tolerances within this space[19,20]. Unexpectedly, a jump in affinity occurred when alkoxyl-linked pyridyl groups such as those present in GSK891(B1) and GSK775(B2) were added. This was puzzling as although

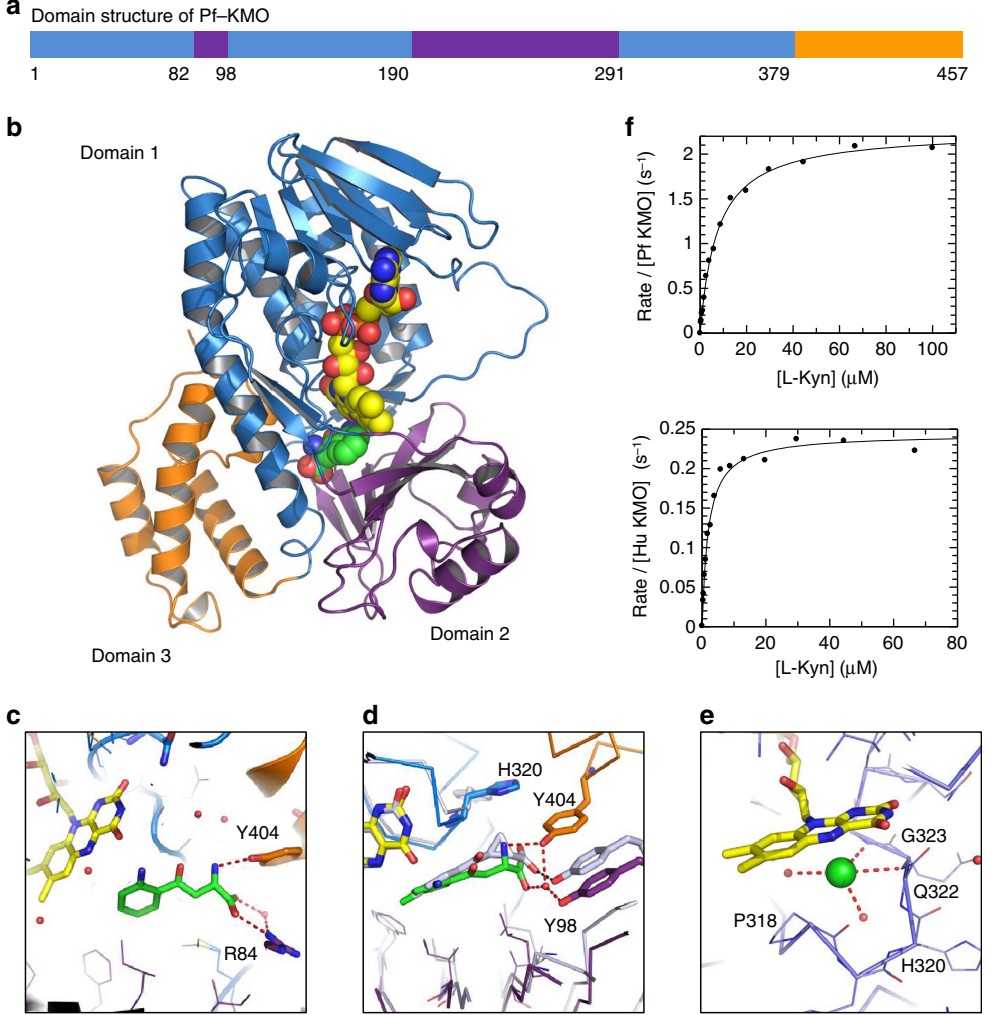

**Figure 2 | Pf-KMO as a structural surrogate for human inhibitor design.** (**a**) Domain structure of Pf-KMO mapped onto a schematic of the primary sequence of the construct crystallized. Domain 1 (blue) is the FAD-binding domain; domain 2 (purple) is a second domain conserved in FPMO enzymes; domain 3 (orange) is a domain unique to KMO. Protein residues retain this domain colouring throughout this figure. (**b**) 1.5 Å X-ray crystal structure of Pf-KMO complexed with substrate, L-Kyn (space fill, carbons in green), FAD (space fill, carbons in yellow). Protein shown in cartoon format. (**c**) Binding of L-Kyn within KMO active site. (**d**) Alternative view of L-Kyn within KMO active site overlaid with *Saccharomyces cerevisiae* KMO-UPF648 inhibitor complex (white carbons, PDB 4J36)[17]. (**e**) Chloride ion (green sphere) bound beneath flavin Pf-KMO, as seen for other family members (Supplementary Fig. 2). (**f**) Determination of L-Kyn $K_M$ values for Pf-KMO and human KMO proteins using a Rapidfire mass spectrometry assay[12]. L-Kyn $K_M$ values of 7 μM for Pf KMO and 2 μM for human KMO were measured in the presence of 200 μM NADPH, with corresponding $k_{cat}$ values of 2.3 and 0.24 s$^{-1}$.

modelling using Pf-KMO had suggested there was sufficient room to accommodate the pendant pyridine, there were no predicted interactions to explain the observed potency increase (Fig. 4e). Our best hypothesis was a possible edge to face interaction with the flavin ring that limits one side of the pocket (Fig. 4f). Intriguingly, initial enzymatic mode of action analysis of series B compounds, such as GSK775(B2), under the same conditions as used for GSK428(A2) suggested non-competitive behaviour (Fig. 4c), leading us to question our assumptions about the binding site and mode of action for these compounds.

**Novel inhibitors have a distinct-binding mode and kinetics.** A structure of GSK775(B2) in Pf-KMO confirmed that it binds within the substrate site (Fig. 5a), with the oxazolidinone core overlapping almost exactly with that in series A compounds. However, the pendant pyridyl group does not reside in the water filled pocket as modelled in Fig. 4e,f. Instead, it π-stacks against a

flavin group that is now tilted away from the substrate site in an orientation not previously observed in KMO, or any other FPMOs we have found (Fig. 5a). In addition to the face to face π interaction with FAD the pyridine also makes edge to face π interactions with Y193 and F238 (Fig. 5b). Flavin movement creates a new solvent channel that opens the active site to the protein surface (Fig. 5c), the pyridine nitrogen of GSK775(B2) H-bonds with an extended water network within this opening (Fig. 5c). R111 needs to move in concert with this flavin tilt, but few other protein changes are necessary to accommodate this movement.

Since GSK428(A2) and GSK775(B2) both occupy the L-Kyn site, the non-competitive behaviour we observed for GSK775(B2) was perplexing, prompting us to undertake a more detailed characterization of the binding kinetics of these compounds (Fig. 5d,e).

First, we characterized the association kinetics by monitoring the onset of inhibition timecourses. For GSK428(A2) timecourses in the presence of inhibitor were linear in the shortest timescale

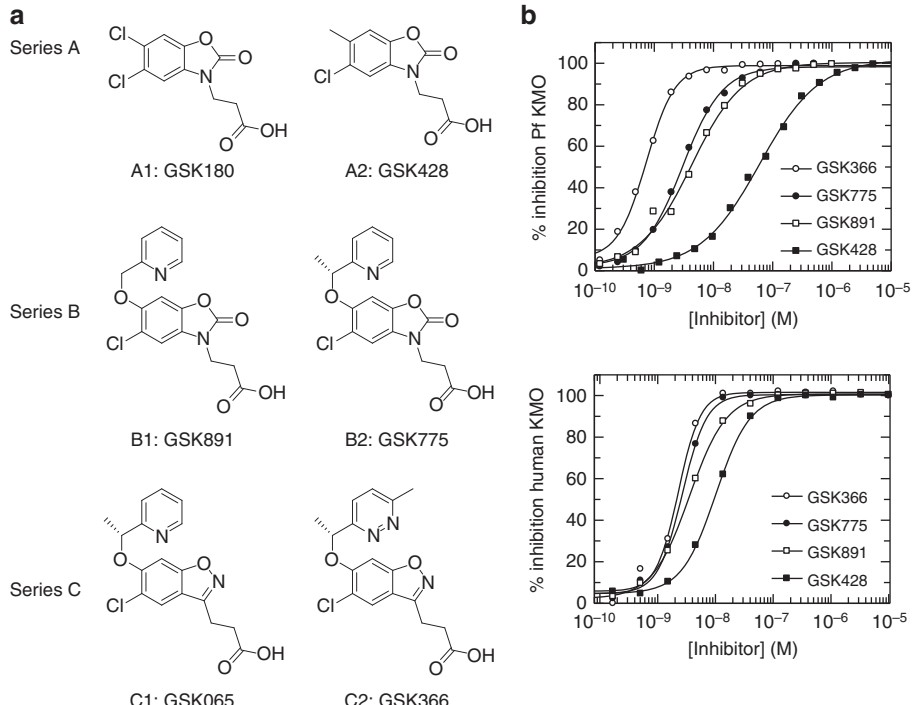

**Figure 3 | Three classes of substrate inspired KMO inhibitors.** These inhibitors show consistent pharmacology for human and *P. fluorescens* enzymes. (**a**) Chemical structures from three inhibitor series (A–C). (**b**) Inhibition curves from members of all three series. Complete inhibition and preserved rank ordering between *P. fluorescens* and human proteins is observed. The steep slopes for some compounds suggest that these inhibitors are at the tight binding limit of the assay. This is consistent with estimates of the concentration of binding-competent KMO from active site titrations[12], which were 1 nM for Pf-KMO and 2 nM for human KMO in these assays.

**Table 1 | IC$_{50}$ values obtained from the inhibition curves shown in Fig. 3b.**

| Compound | IC$_{50}$ Human KMO | IC$_{50}$ Pf-KMO |
|---|---|---|
| GSK366 | 2.3 nM | 0.7 nM |
| GSK775 | 2.7 nM | 3.0 nM |
| GSK891 | 3.6 nM | 4.3 nM |
| GSK428 | 10.0 nM | 60.0 nM |

that could be achieved in manual mixing experiments, indicative of rapid association on this timescale with $k_{obs}$ greater than 0.6 min$^{-1}$ under these conditions (Supplementary Fig. 4a). However for GSK775(B2), under the same concentration conditions, clear curvature in the presence of inhibitor was seen indicative of slow onset of inhibition (Fig. 5d). Exponential analysis gave a $k_{obs}$ value of 0.062 min$^{-1}$, at least 10 times slower than for GSK428(A2). Dissociation kinetics were characterized by pre-equilibrating enzyme with an excess of inhibitor, and initiating reactions by adding a large excess of L-Kyn relative to $K_M$ to drive maximum recovery of activity. Figure 5e shows the timecourse of recovery of activity after adding substrate. For GSK428(A2) no evidence of a lag in recovery of enzyme activity was seen in the shortest timescale that could be achieved in manual mixing experiments, indicative of rapid dissociation on this timescale, with $k_{obs}$ greater than 1.0 min$^{-1}$ (Supplementary Fig. 4b). For GSK775(B2) a lag phase was clearly observed indicative of slow dissociation on this experimental timescale, with $k_{obs} = 0.006$ min$^{-1}$.

Taken together these data suggest a differentiated kinetic profile for GSK775(B2) and GSK428(A2), with GSK775(B2) showing slow onset of inhibition and slow dissociation, in contrast to the rapid kinetics of GSK428(A2). The apparently

non-competitive behaviour of GSK775(B2) can be rationalized by the unexpected slow dissociation of the prebound inhibitor relative to the timescale of the initial competition assay (~2 h half life for the KMO-GSK775(B2) complex).

**Tilting inhibitors designed for potency and residence times.** To further improve the pharmaceutical properties of our inhibitors and to investigate whether inducing a tilting flavin could be a general tactic used to deliver high potency and long residence times for this important enzyme, we used the KMO-GSK775(B2) crystal structure to design molecules that differed in both the heterocyclic core and the tilt inducing pendant group[20].

Grafting the 1-(pyridin-2-ylethoxy) pendant group of GSK775(B2) onto the benzisoxazole core gave GSK065(C1) which preserves the tilting flavin and maintains the high *in vitro* activity of GSK775(B2) (IC$_{50} = 2.5$ nM, $n = 12$). A variety of alternative 'tilting' pendant substituents were also incorporated onto the benzisoxazole ring, as exemplified by the methyl-pyridazine ring of GSK366(C2). This aromatic ring retains the tilting flavin conformation (Fig. 6a), as expected, and induces an even clearer network of water interactions within the new solvent channel (Fig. 6b). Pyridazine has been promoted as an excellent bioisostere for pyridine, often with the ability to enhance solubility and decrease metabolic liability[21]. The mean potencies of GSK366(C2) and GSK065(C1), 2 nM ($n = 4$) and 4.5 nM ($n = 7$) respectively, are of the same order as the concentration of human enzyme added to the reaction and the slopes of their inhibition curves are steep for both compounds; both have a mean slope of 1.9. This suggests the IC$_{50}$ values quoted may underestimate their true affinities. Indeed, detailed kinetic analysis for GSK065(C1) and GSK366(C2) (Fig. 6c,d, Supplementary Figs 5 and 6) suggest that their true $K_i$s may be around 50 pM and 12 pM, respectively. The slow onset of

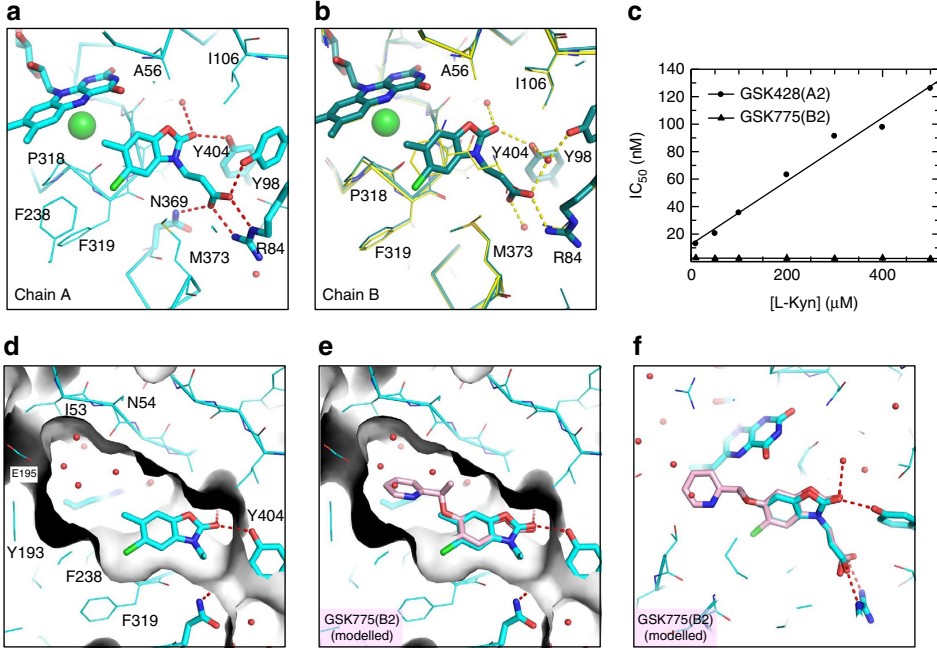

**Figure 4 | Structural and enzymatic characterization of GSK428(A2) and GSK775(B2). (a)** Binding mode of GSK428(A2) (cyan) in active site of chain A of the ASU of a 1.4 Å crystal structure. Key H-bonds and salt bridges shown. **(b)** Analogous view of GSK428(A2) molecule in B chain of ASU. Subtle variations of interactions, between the two chains are apparent, for example, monodentate salt bridge with R84. To illustrate relative position of native substrate, Pf-KMO-L-Kyn complex structure is shown in yellow **(c)** Effect of [L-Kyn] on $IC_{50}$ of GSK428(A2) and GSK775(B2) for human KMO. A linear relationship, as seen for GSK428(A2), is typical of simple Kyn-competitive inhibitors. The behaviour of GSK775(B2), which shows no dependence on [L-Kyn], is expected of non-competitive inhibitors. **(d)** Active site with GSK428(A2) bound (chain A) and surface representation (grey) of protein to highlight water filled pocket used for inhibitor design **(e,f)** Modelled structure of GSK775(B2) (pink) preserving the position of the oxazolidinone core and placing the pendant pyridine into this cavity, overlaid with GSK428(A2) structure. The close proximity of the pyridine to the flavin is apparent.

inhibition and slow dissociation from the KMO enzyme for series C compounds are readily apparent, with GSK366(C2) exhibiting a dissociation half life of the order of 12 h (Fig. 6d).

The benzisoxazole series, exemplified here by GSK065(C1) and GSK366(C2), illustrates the ability to translate our original observation of tilting flavin and slow kinetics to successfully design new inhibitors that preserve this unusual-binding mode and desirable kinetic signature[22–24].

**Type II inhibition does not stimulate peroxide generation.** Flavin dynamics during the catalytic cycle of FPMO class A enzymes has been a subject of considerable interest, as orchestrated motions potentially govern substrate binding and product release as well as being critical for regulation of the reductive and oxidative steps of the catalytic process. A range of flavin conformations has been captured by structural studies, leading to terms such as 'waving'[25–27] and 'flapping'[28] flavin to describe the repertoire of motions observed. The tilting flavin orientation seen for series B and C inhibitors does not match any of those reported and is much more appropriately described as a tilt, as it lacks the lateral motion for the wave that leads to the 'out' state first reported in the class A exemplar enzyme PHBH (Fig. 7a).

In PHBH, flavin movement from the *in* to the *out* state is governed by the protonation state of the phenolic substrate, with only the *out* flavin believed to be accessible for hydride transfer from a transiently bound NADPH molecule[29]. Once reduced, the flavin swings back into the enclosed active site and sits in the *in* conformation ready for incorporation of molecular oxygen to form the reactive C4a-hydroperoxide intermediate, which then hydroxylates the prebound substrate. This acts as a control point to ensure efficient coupling of reductive and oxidative steps in the

catalytic cycle, with the hydroperoxy flavin only produced when the substrate is present to accept its hydroxy modification. Dissociation of substrate before hydroxylation results in the uncoupled production of hydrogen peroxide. Unactivated substrate mimetics such as NBA and BA are KMO inhibitors known to stimulate peroxide ($H_2O_2$) production as the catalytic cycle is frustrated at the hydroxylation step and the system cycles by eliminating hydrogen peroxide[9] and we also observe similar behaviour with GSK428(A2) using an HRP/Amplex Red assay to detect peroxide formation (Fig. 8). However, inhibitors from series B and C did not stimulate $H_2O_2$ production, and in the human enzyme there was even an apparent reduction of $H_2O_2$ production below basal levels (Fig. 8a). Given this distinctive mechanism and binding mode we decided to name this mechanism of inhibition type II, to distinguish it from the simple substrate mimicry that stimulates $H_2O_2$ production (type I inhibitor).

We hypothesized that the unique tilting flavin conformation induced by our series B and C inhibitors may interrupt the catalytic cycle of KMO at an alternative junction to simple substrate mimetics, trapping the isoalloxazine ring in a limbo state unable to the complete the trajectory required to be reduced by NADPH or form the hydroperoxo species. Curious to provide greater mechanistic rationale for this distinction we attempted to co-crystallize substrate and inhibitors in the presence of NADPH as well as with a small nicotinamide molecule, all without success. While it is assumed that simultaneous binding of FAD-NADPH-substrate is a requisite for 'cautious', class A, FPMO regulation[30], ternary X-ray complexes are scarce. The only known NADPH bound complex of a class A enzyme, is that of a PHBH Variant, R220Q (PDB ID:1KOJ[26]). Surprisingly this captures the NADPH within a surface groove in an extended conformation where the

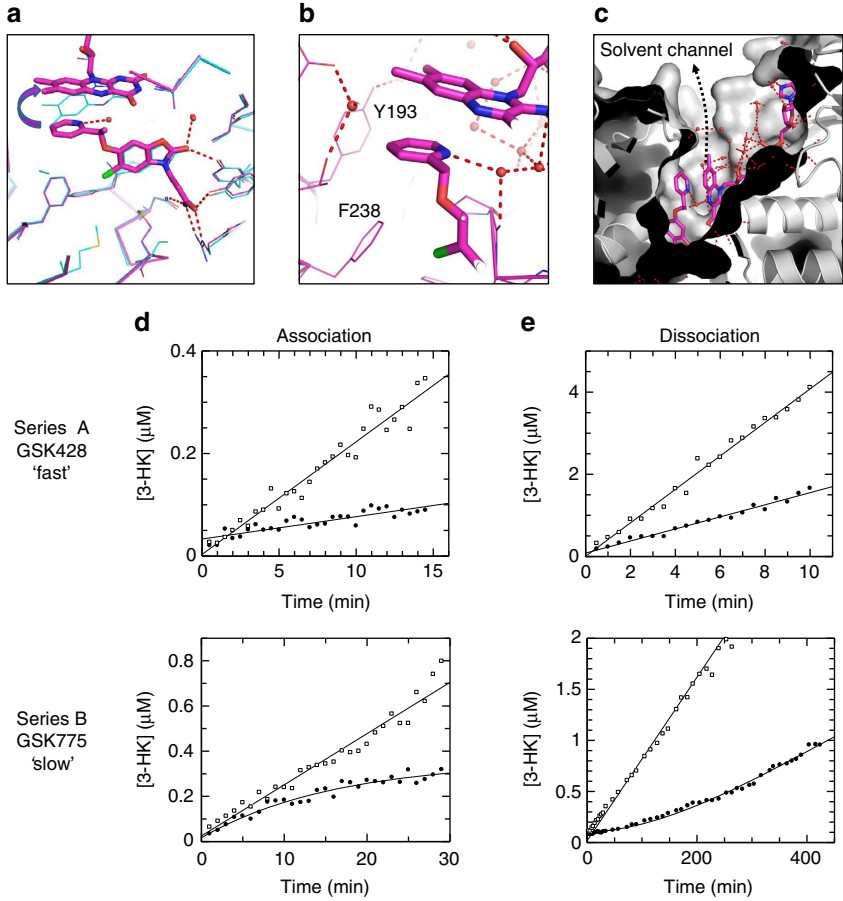

**Figure 5 | Differential structural and kinetic signatures of series A and B. (a)** Crystal structure of GSK775(B2) bound in Pf-KMO (magenta) overlaid with that of GSK428(A2) (cyan). Tilting movement of the flavin, away from pyridine group shown by arrow. H-bonds of oxazolidinone core are maintained by both inhibitors (red). **(b)** Environment around GSK775(B2) pyridine. H-bonding with water network now accessible due to flavin movement, $\pi$-stacking interactions with flavin. **(c)** Movement of flavin creates a solvent filled channel from active site to surface of the protein. **(d)** Onset of inhibition for GSK428(A2) and GSK775(B2) for human KMO. Uninhibited timecourse (open symbols), onset of inhibition timecourse in the presence of 10 nM inhibitor (closed symbols). No evidence of slow binding was seen for GSK428(A2) on this timescale. GSK775(B2) showed slow onset of inhibition with $k_{obs} = 0.062\,min^{-1}$. **(e)** Recovery of activity for GSK428(A2) and GSK775(B2) for human KMO. Uninhibited timecourse (open symbols), recovery of activity timecourses (closed symbols). No evidence of slow recovery of activity was seen for GSK428(A2) on this timescale. GSK775(B2) showed slow recovery of activity with $k_{obs} = 0.006\,min^{-1}$.

nicotinamide extends away from, rather than towards active site[31] suggesting this may be an unproductive-binding mode. Mutagenesis studies do suggest residues that line this surface channel (R33, Y38, Y42, Y44, F161, R166) are involved in NADPH binding, but spectroscopic data are more consistent with a reverse orientation that more simply brings the nicotinamide ring close to the flavin in its *out* conformation. Many of these residues lie within the solvent channel identified as accessible from the active site in our tilting flavin structures, enabling us to model a NADPH molecule along this channel to place the nicotinamide ring on the *re* face of the flavin. Interestingly, this ring position closely resembles that found in the catalytically competent FAD-NADPH-substrate complexes of class B monooxygenases[28,32,33] (Fig. 7c,d, Supplementary Fig. 7).

Trapping class B, 'bold', monooxygenases, during the catalytic cycle has been more successful. In these sister enzymes reduction of FAD by NADPH occurs equally well in the presence or absence of bound substrate. To protect against uncoupling of the reductive and oxidative half cycles, the spent NADP(H) cofactor remains bound to the protein throughout the reaction cycle, protecting both the reduced and the C4a-hydroperoxide species from being quenched[30]. Dramatic changes in the flavin include a

flipped orientation (Fig. 7c) observed along a catalytic trajectory captured in a remarkable series of crystal structures of the class B hydroxylase, KtzI[28], that traps reduced, oxidized and various substrate and cofactor bound states. Transition between the *in* and a *flipped* flavin orientation requires a 180° rotation of the isoalloxazine ring, leading to the idea that a 'flapping' flavin motion may be required for this enzyme[28]. Importantly, these snapshots include those in the presence of NADP(H) providing an invaluable window into the choreographed motions of flavin and nicotinamide rings within the reaction cycle.

Our modelled ternary KMO complex suggests the degree of flavin motion could also be subtly divergent amongst class A enzymes, with a tilting flavin creating sufficient space to accommodate NADPH binding (Fig. 7d). This close arrangement of NADPH, FAD and substrate may account for the relatively low $K_M$ measured for NADPH for human KMO[12]. The model suggests the nicotinamide ring could get close enough to occupy the same space as the pendant group of our type II inhibitors. The observed rate of inhibitor binding is not diminished in response to increasing NADPH concentrations (Supplementary Fig. 9), which suggests that the inhibitor and NADPH bind to different states of the enzyme, but this does not exclude the possibility that

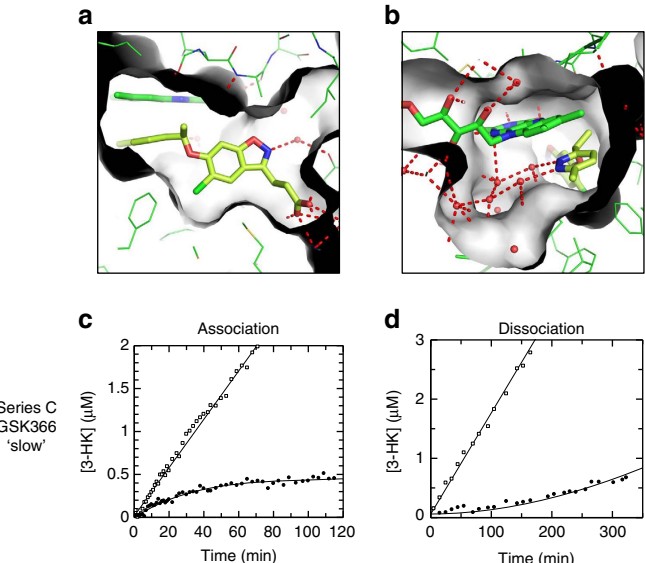

**Figure 6 | The benzisoxazole inhibitor series maintain the tilting flavin orientation that retains slow kinetics.** Panels **a**–**d** show the characterization of the representative series C inhibitor, GSK366(C2). (**a**,**b**) GSK366(C2) binds in the active site and tilts the flavin to access and make even more extensive interactions with the new water filled channel. (**c**) Onset of inhibition for GSK366(C2) for human KMO. Uninhibited timecourse (open symbols), onset of inhibition timecourse in the presence of 5 nM GSK366(C2) (closed symbols). Slow onset of inhibition was observed with $k_{obs} = 0.031\,min^{-1}$. (**d**) Recovery of activity timecourse for GSK366(C2). Uninhibited timecourse (open symbols), recovery of activity timecourse (closed symbols). Slow recovery of activity was observed with $k_{obs} = 0.001\,min^{-1}$.

type II inhibitors may change the ability of NADPH to bind to the inhibited enzyme, or do so in a productive manner. For example, they may inhibit the reductive step consistent with the tilting flavin being a high energy conformation induced only by type II inhibitor binding, or perhaps a transient state along the catalytic trajectory pinned by the inhibitor in a conformation that is unable to be reduced by NADPH binding or oxidized by molecular oxygen (Fig. 8b).

In summary, type II inhibitors may exert three concerted but separate effects: first, prevent productive association with NADPH, killing the reductive half-reaction; second, prevent substrate binding, killing the oxidative half-reaction; and finally, pushing the FAD into an unproductive position. The balance of these contributions may vary amongst inhibitors. Therefore, type II KMO inhibitors realize the ideal profile of preventing 3-HK generation without peroxide generation and provide a structural template of how this can be achieved. This represents a major step in drug design for KMO.

## Discussion

KMO is an important metabolic enzyme and drug target for which structural insights have so far been remarkably limited. While the human enzyme has been recalcitrant to structural studies a double mutant of Pf-KMO has enabled the first substrate bound structure to be determined. Success was made possible by the use of a full-length *P. fluorescens* orthologue that included not only the FAD and substrate-binding domains common to all FPMOs but also a third C-terminal domain that is unique to the KMO enzyme. This inclusion proved critical to full understanding of substrate site recognition, as the novel helical domain is an integral part of the substrate-binding site. The

KMO-L-Kyn complex brings new insights into substrate binding and challenges dogma in the design of KMO inhibitors. For example, it has long been assumed essential to include a hydrogen bond acceptor to mimic key interactions made by the carbonyl of L-Kyn, yet in our structure the carbonyl makes no significant interactions. This crystallographic system therefore represents a significant advance in our ability to apply structure-based design to inhibitors of this enzyme, allowing us to address puzzling structure–activity relationships and providing a firm basis for novel inhibitor discovery and mechanistic analysis.

This promise has already been realized with our discovery of the first type II KMO inhibitors that bind to an unprecedented tilting flavin conformation. Crystallographic complexes of our extended oxazolidinone inhibitor series (B) rationalize their unexpectedly high potency, by demonstrating that they bind with this intriguing variation of the flavin position. Unlike previous substrate site inhibitors that carry the mechanistic liability of stimulated hydrogen peroxide production, type II compounds prevent formation of the flavin peroxo species and do not possess this undesirable side effect. Trapping the flavin in an unproductive orientation decouples substrate site binding from NAPDH reduction and flavin oxidation and is associated with increased inhibitor residence times. These type II inhibitors therefore have all the hallmarks of ideal KMO inhibitors. Our understanding of their structural and mechanistic basis has enabled us to translate these ideal attributes to distinct chemical series as exemplified by the optimized type II benzisoxazole compounds GSK065(C1) and GSK366(C2). These have the full profile of pharmaceutical and pharmacological *in vitro* and *in vivo* properties that make them suitable candidates for progression to clinical evaluation in acute pancreatitis multiple organ dysfunction syndrome[20].

Many FAD-dependent monooxygenases depend on carefully choreographed flavin motions for catalysis. While we are the first to report type II inhibitors for FPMOs, trapping the flavin in unproductive positions may be a general approach applicable to a broad range of FPMOs, opening the doors to rich opportunities for innovative inhibitor design.

## Methods

**Protein expression and purification.** For the structural studies of apo and inhibitor-bound Pf-KMO, the expression vector pET17b Pf-KMO C252S/C461S[34] was transformed into *Escherichia coli* BL21 (DE3). Cells were plated onto LB agar plates containing 100 µg ml$^{-1}$ ampicillin and incubated for 16 h at 37 °C. Plates were scraped into 200 ml LB broth and used for the inoculation of 12 × 11 of LB broth. The culture was grown for 12–16 h (until OD$_{600}$ 2.7) at 22 °C. After the agar plate stage no ampicillin was used, and no IPTG was used for induction. Cells were collected by centrifugation, chilled to 4 °C and frozen at −80 °C.

Cells were resuspended in 20 mM Hepes pH 7.5 containing 1 mM DTT, 100 µM FAD and protease inhibitor cocktail (100 µM AEBSF, 80 nM aprotinin, 5 µM bestatin, 1.5 µM E-64, 2 µM leupeptin, 1 µM pepstatin A). The cell suspension was lysed by sonication with ice cooling. The suspension was centrifuged at 30,750g for 30 min at 4 °C. Streptomycin sulfate was added to the supernatant over 15 min to a final concentration of 1.5% w/v. The suspension was centrifuged at 19,700g for 10 min. The supernatant was slowly brought to 50% ammonium sulfate saturation over 20 min at 4 °C. The suspension was centrifuged at 19,700g for 10 min. The pellet was dissolved in 20 mM Hepes pH 7.5, 1 mM DTT and the solution was frozen at −80 °C before being further purified.

A 15Q Source ion exchange column was equilibrated with 20 mM Hepes pH 7.5, 10 mM NaCl, 1 mM DTT at 5 ml min$^{-1}$. The redissolved ammonium sulfate pellet from above was loaded to the column at 4 ml min$^{-1}$ followed by extensive washing with buffer at 5 ml min$^{-1}$. The column was eluted at 5 ml min$^{-1}$ with a 20 min gradient to 20 mM Hepes pH 7.5, 500 mM NaCl, 1 mM DTT. The Pf-KMO containing peak was further purified by size exclusion chromatography on a HiLoad 26/60 Superdex 200 column equilibrated in 20 mM Hepes pH 7.5, 10 mM NaCl, 1 mM DTT run at 1 ml min$^{-1}$. Fractions containing pure Pf-KMO were pooled, supplemented with 5% glycerol, concentrated to 18 mg ml$^{-1}$ using an Amicon 10,000 molecular weight cut-off centrifugal concentrator and frozen at −80 °C in aliquots.

For the expression and isolation of human KMO membranes, full-length human KMO was cloned as an N-terminal GST fusion, expressed in Sf9 cells and isolated as a membrane fraction as described in Lowe *et al.*[12]

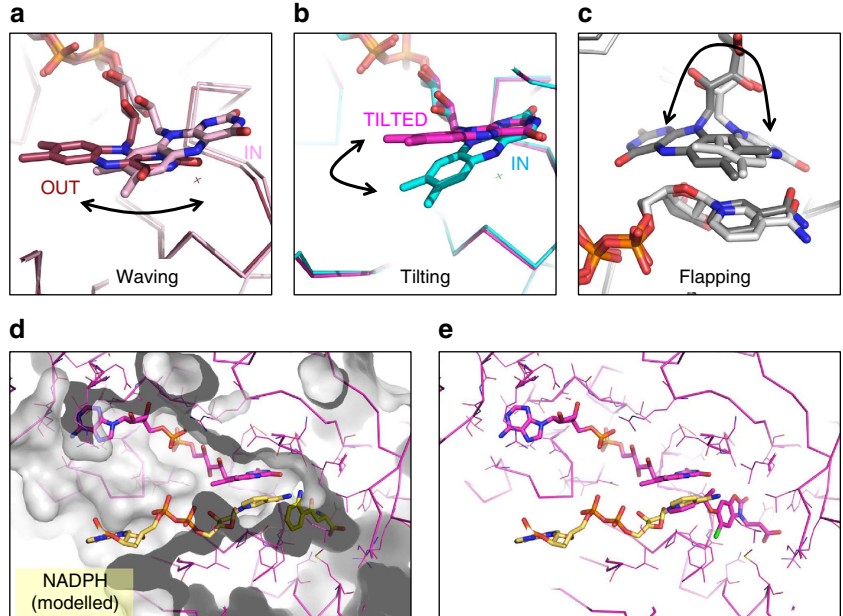

**Figure 7 | Repertoire of flavin motion and modelling NADPH binding.** (**a**) Overlay of two PHBH enzyme complexes that depict in (pink, PDB 1DOC) and out (brown, PDB 1DOD) positions characteristic of 'Waving' flavin movement[27] that pivots around the ribityl C(2) carbon. (**b**) Overlay of two Pf-KMO inhibitor structures that illustrate in (cyan, GSK428(A2)) and tilted flavin positions which hinges on the two carbonyls of the pyrimidinone flavin ring. (**c**) Overlay of two class B KtzI N-hydroxylase complexes showing in (white, KtzI − FADred − NADP$^+$ − L-Orn, PDB 4TLX) and 'flapped' (grey, KtzI − FADred-ox − NADP$^+$ − L-Orn, PDB 4TM0) positions. Reorientation of isoalloxazine ring between two positions has been described as 'flapping' as this requires a 180 degree rotation. (**d**) NADPH molecule is positioned into the solvent channel and active site of the KMO-FAD-GSK775(B2) complex with the tilting flavin. The GSK775(B2) molecule has been removed and instead L-Kyn is positioned within the active site, guided by the KMO-FAD-L-Kyn complex. NADPH is positioned to ensure there are no severe atom clashes and maintain an appropriate geometry. No attempts were made to allow protein sidechain movement or optimize interactions. (**e**) Same model as (**d**) with GSK775(B2) rather than L-Kyn preserved in the active site. The nicotinamide ring of NADPH occurs in the same space as the GSK775(B2) pyridine.

For the structural studies of the L-Kyn Pf-KMO complex, the expression vector pET17b Pf-KMO C252S/C461S[34] was transformed into *E. coli* BL21 (DE3). Cells were added to 10 × 10 ml LB broth containing 100 µg ml$^{-1}$ ampicillin and incubated for 16 h at 37 °C with shaking at 200 r.p.m. The 10 ml starter growths were added to 10 × 1 l of LB broth containing 25 µg ml$^{-1}$ ampicillin. The culture was grown for 12–16 h at 37 °C with shaking at 180 r.p.m., no IPTG was used for induction. Cells were collected by centrifugation, chilled to 4 °C and frozen at − 20 °C.

Cells were resuspended in 20 mM Hepes pH 7.5 containing 10 mM NaCl, 1 mM DTT and 200 µM PMSF. The cell suspension was lysed by sonication with ice cooling. The suspension was centrifuged at 47,800*g* for 60 min at 4 °C. The supernatant was filtered through a 0.45 µm MF-Millipore membrane and loaded onto a 60 ml FPLC Q Sepharose 26/10 column pre-equilibrated with 20 mM Hepes pH 7.5, 10 mM NaCl, 1 mM DTT at 4 ml min$^{-1}$. The column was then washed with 400 ml 20 mM Hepes pH 7.5, 50 mM NaCl, 1 mM DTT at 4 ml min$^{-1}$. The column was eluted with 500 ml 20 mM Hepes pH 7.5, 110 mM NaCl, 1 mM DTT at 4 ml min$^{-1}$. The pooled fractions of Pf-KMO was slowly brought to 50% ammonium sulfate saturation over 20 min on ice. The suspension was centrifuged at 38,720*g* for 20 min at 4 °C. The pellet was frozen at − 20 °C before being further purified.

The ammonium sulfate pellet was dissolved in 20 mM Hepes pH 6.8 containing 10 mM NaCl and 1 mM DTT. This solution was concentrated by centrifugation using VIVASPIN MW 30,000 concentrators to a volume of 2 ml. This was loaded onto a 320 ml 26/600 Superdex 75 column equilibrated with 20 mM Hepes pH 6.8, 10 mM NaCl, 1 mM DTT and run at 1 ml min$^{-1}$. Fractions containing pure Pf-KMO were pooled, supplemented with 5% glycerol, and concentrated by centrifugation using VIVASPIN MW 30,000 concentrators to 32 mg ml$^{-1}$. Aliquots were flash frozen in liquid nitrogen and stored at − 80 °C.

**Crystallization and structure determination.** To determine the structure of apo and inhibitor-bound KMO, crystals of *P. fluorescens* KMO were grown at 4 °C by sitting drop vapour diffusion using a protein concentration of 18 mg ml$^{-1}$, with crystallization solutions consisting of either 20% glycerol, 14.4% PEG 8000, 0.08 M sodium cacodylate pH 6.5, 0.16 M calcium acetate or 20% glycerol, 16% PEG 8000, 0.08 M sodium cacodylate pH 6.5, 0.16 M magnesium acetate. Crystallization drops were set-up using 100 nl protein solution mixed with 100 nl crystallization solution (or 200 nl + 200 nl) above a reservoir of 40 µl of the same crystallization solution.

To make the crystal soaking solutions 1 µl of a 500 mM compound stock solution in DMSO was added to 19 µl of the crystallization solution to give a compound concentration of ∼ 25 mM (5% DMSO). The crystal soaking solution was added to the crystallization drop to excess and left at 4 °C for 3–8 h. Crystals were then frozen in liquid nitrogen directly from the soaking drops, without any additional cryo-protectant.

X-ray diffraction data were collected at 100 K at the European Synchrotron Radiation Facility (ESRF). The data were processed and scaled using autoPROC[35] or the ESRF automated beam line data processing software, utilizing XDS[36], AIMLESS[37] and the CCP4 suite of programs[38]. The crystal space group is P2₁ with unit cell dimensions approximately $a = 69.7$ Å, $b = 53.1$ Å, $c = 136.5$ Å, $\alpha = 90°$, $\beta = 103.8°$, $\gamma = 90°$ and two protein molecules in the asymmetric unit. Data collection statistics (Supplementary Table 1).

The structures were determined using the coordinates of an isomorphous unliganded protein model, with preliminary refinement carried out using autoBUSTER[39]. In all cases, the ligands were clearly visible in the resulting $F_o − F_c$ electron density maps (Supplementary Fig. 8). Coot[40] was used for model building, with refinement completed using autoBUSTER. The statistics for the final models are given in Supplementary Table 1. The coordinates and structure factors have been deposited in the Protein Data Bank.

For the structural determination of L-Kyn substrate complex, crystals of *P. fluorescens* KMO were grown at 18 °C by hanging drop vapour diffusion using a protein concentration of 32 mg ml$^{-1}$, with crystallization solutions consisting of 20% glycerol, a range of 11–19% PEG 8000, 0.08 M sodium cacodylate pH ranging from 5.5 to 7.0 and 0.16 M calcium acetate. Crystallization drops were set-up using 2 µl protein solution mixed with 2 µl crystallization solution above a reservoir of 500 µl of the same crystallization solution.

Crystal soaking solution contained 20% glycerol, 25% PEG 8000, 0.08 M sodium cacodylate of pH relevant to the crystallization solution, 0.16 M calcium acetate and 1 mM L-Kyn. Crystals were extracted from the crystallization solution and placed in a 1 µl drop of crystal soaking solution. The well was re-sealed and left for roughly 48 h. Crystals were then frozen in liquid nitrogen, without any additional cryo-protectant.

X-ray diffraction data were collected at 100 K at the Diamond Light Source. The data were processed and scaled using iMosflm[41] the CCP4 suite of programs[38]. The crystal space group is P2₁ with unit cell dimensions approximately $a = 69.9$ Å, $b = 52.6$ Å, $c = 138.0$ Å, $\alpha = 90°$, $\beta = 104.0°$, $\gamma = 90°$ and two protein molecules in the asymmetric unit. Data collection statistics (Supplementary Table 1).

The structures were determined using the coordinates of an isomorphous unliganded protein model, with preliminary refinement carried out using Refmac5

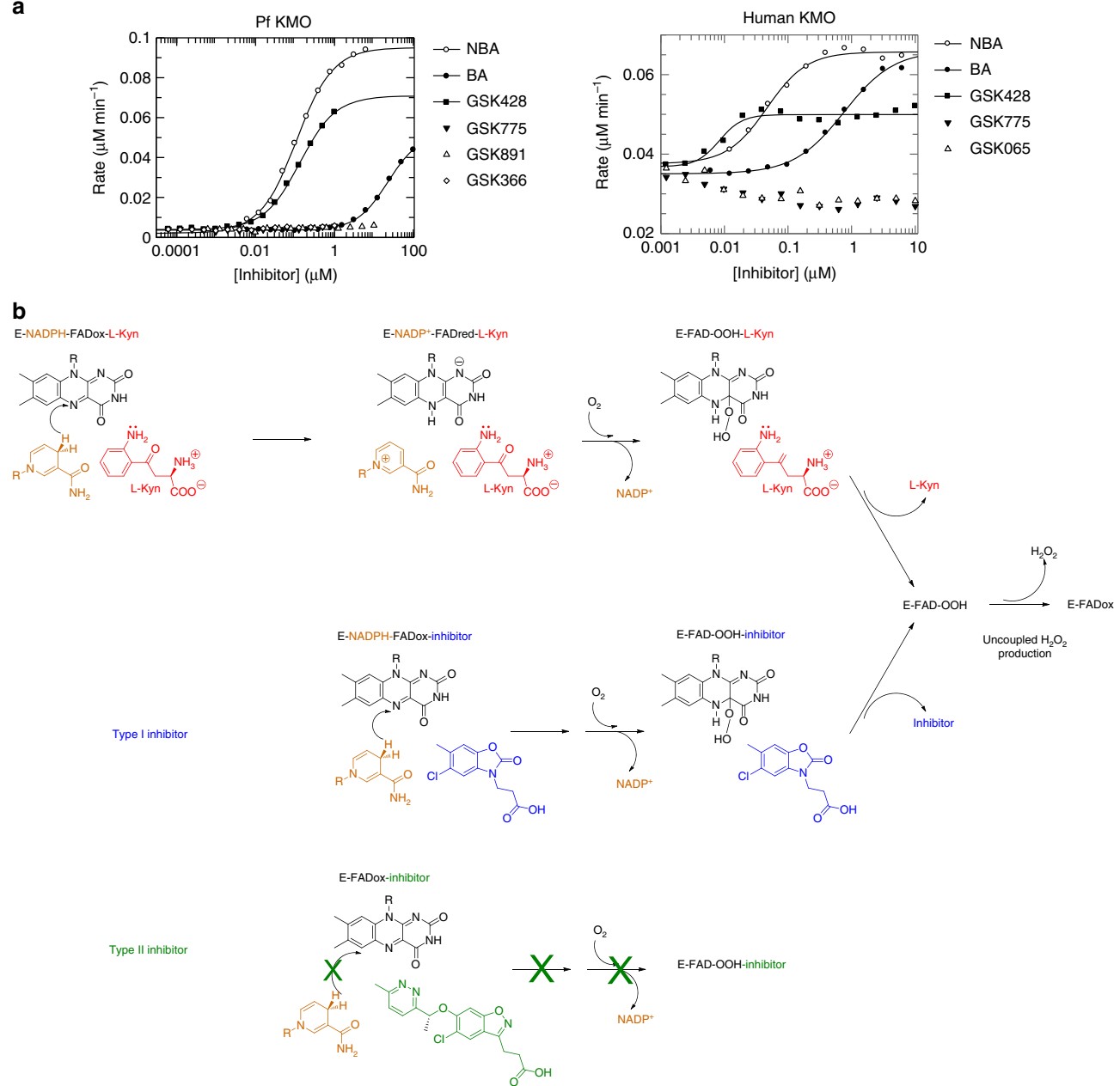

**Figure 8 | Type II inhibitors do not promote peroxide generation.** (**a**) Generation of hydrogen peroxide by Pf-KMO and human KMO in the presence of different inhibitors was monitored in a HRP/Amplex Red assay. Classical (NBA = m-nitrobenzoylalanine; BA = benzoylalanine) and series A compounds showed a dose dependent stimulation of $H_2O_2$. No stimulation was observed for series B and C inhibitors. (**b**) Schematic of the reaction steps that lead to $H_2O_2$ formation in the presence of substrate and type I inhibitors such as GSK428(A2). Type II tilting inhibitors such as GSK366(C2), may inhibit at multiple steps including productive NADPH association, substrate binding and formation of the FAD hydroperoxy species.

(ref. 42). In all cases, the ligands were clearly visible in the resulting $F_o − F_c$ electron density maps (Supplementary Figure). Coot[40] was used for model building, with refinement completed using Refmac5. The statistics for the final models are given in Supplementary Table 1. The coordinates and structure factors will be deposited in the Protein Data Bank on acceptance of this article.

**Enzymatic activity assays and inhibitor-binding kinetics.** Inhibitor $IC_{50}$ values against isolated human KMO were determined using a Rapidfire mass spectrometry assay as described in Lowe *et al.*[12] utilizing the Sf9 cell expressed human KMO described above. The inhibitor $IC_{50}$ values against Pf-KMO were determined using an analogous Rapidfire mass spectrometry assay, utilizing the purified protein described above. Final assay concentrations were [Pf-KMO] = 1 nM, [L-Kyn] = 10 μM and [NADPH] = 200 μM.

The $K_M$ of kynurenine for human KMO and for Pf KMO were measured as described in Lowe *et al.*[12]. The $K_M$ of NADPH for human KMO was measured

using a regeneration system as described in Lowe *et al.*[12], with a kynurenine concentration of 10 μM.

For the association kinetics assays, measurements were performed in a buffer of 50 mM Hepes pH 7.5, 2 mM DTT, 1 mM EDTA, 0.1 mM Chaps. Reactions were performed at ambient temperature, by addition of human KMO enzyme to a mixture of L-Kyn and NADPH containing the test inhibitor. Concentrations of components after initiation were 8 μg ml$^{-1}$ KMO membranes (∼2 nM active concentration by active site titration), 200 μM NADPH, 10 μM L-Kyn and the stated concentration of inhibitor. Aliquots of the mixtures were quenched into 0.5% TFA in water. L-Kyn and 3-HK levels in the quenched samples were measured using the Rapidfire mass spectrometry method described in Lowe *et al.*[12] Calibration curves for L-Kyn and 3-HK were used to correct for the lower detection response for 3-HK. Per cent turnover and resulting concentration of 3-HK were calculated for each time point in Excel. $k_{obs}$ values for onset of inhibition were obtained by fitting individual timecourses to equation 1 where $t$ = time, $v_s$ = steady-state rate, $v_i$ = initial rate and $c$ = background[43]. $v_i$ was fixed at the

uninhibited rate obtained from the slope of the timecourse in the absence of inhibitor.

$$[3HK] = v_s t + \frac{v_i - v_s}{k_{obs}} \left[ 1 - e^{-k_{obs}t} \right] + c \qquad (1)$$

For the full kinetic analysis of GSK065(C1) and GSK366(C2), association timecourses were measured at multiple inhibitor concentrations and the association rate constant ($k_{on}$) was obtained from the gradient of a plot of $k_{obs}$ against [inhibitor].

For the Association kinetics experiments of GSK775(B2) at sub-saturating NADPH concentrations, measurements were performed as described above, in the presence of an NADPH regeneration system comprising D-glucose-6-phosphate and glucose-6-phosphate dehydrogenase. Concentrations of components after initiation were 10 nM GSK775(B2), 32 µg ml$^{-1}$ KMO membranes ($\sim 8$ nM active concentration by active site titration), 10 µM L-Kyn, 3 mM D-glucose-6-phosphate, 1 unit ml$^{-1}$ glucose-6-phosphate dehydrogenase and the stated concentration of NADPH. $k_{obs}$ values for onset of inhibition were obtained by fitting individual timecourses to Equation (2), describing onset of inhibition under non pseudo-first order conditions[43].

The dissociation kinetics measurements were performed in a buffer of 50 mM Hepes pH 7.5, 2 mM DTT, 1 mM EDTA, 0.1 mM Chaps. Mixtures of human KMO enzyme and excess inhibitor were pre-equilibrated for 1 h at ambient temperature, after which reactions were initiated by the addition of a high concentration of L-Kyn. Concentrations of components after initiation were 8 µg ml$^{-1}$ KMO membranes, 200 µM NADPH, 500 µM L-Kyn and the stated concentration of inhibitor. Aliquots of the reaction mixtures were quenched into 0.5% TFA in water. 3-HK levels in the quenched samples were measured using the Rapidfire mass spectrometry assay described in Lowe et al.[12] A calibration curve for 3-HK was used to convert the data to concentration of 3-HK at each time point. $k_{obs}$ values for recovery of activity were obtained by fitting individual timecourses to equation 1 with $v_i$ (initial rate) set to zero.

For the full kinetic analysis of GSK065(C1) and GSK366(C2), timecourses in the presence and absence of inhibitor were simultaneously fitted to a kinetic scheme of competitive inhibition using Kintek v 5.2.16 (refs 44,45). The scheme assumed rapid equilibrium substrate binding with L-Kyn $K_D = K_M$ and an irreversible hydroxylation step[9]. In this analysis, inhibitor $k_{on}$ was fixed at the value obtained in the association kinetic studies, and $k_{cat}$ and inhibitor $k_{off}$ were the two variable parameters.

**Measurement of hydrogen peroxide formation.** Measurements were performed in a buffer of 50 mM Hepes pH 7.5, 0.1 mM Chaps. A two-fold dilution series of test inhibitor and control compounds was prepared from DMSO solutions using the Hewlett-Packard D300 digital dispenser in a 384-well plate and normalized to a final DMSO concentration of 1% in the assay. 10 µl of 80 µg ml$^{-1}$ human KMO or 40 nM Pf-KMO were added. After 15 min incubation, 10 µl of a substrate/detection mix (2 U ml$^{-1}$ HRP, 20 µM Amplex Red, 40 µM NADPH) were added and the plate was read kinetically on a Tecan M1000 fluorescent plate reader with excitation wavelength 530 nm and emission wavelength 580 nm. A calibration curve comprising a two-fold dilution series of resorufin from 20 µM top concentration was included on the plate, and used to convert assay data to concentration of resorufin formed. Initial rate (µM min$^{-1}$) was extracted by linear fitting of the early region of each timecourse. Dose responses were fitted to a four parameter logistic expression where appropriate.

**Data availability.** Data supporting the findings of this study are available within the article and its Supplementary Information files and from the corresponding author upon reasonable request. The final models of the crystal structures presented within this paper have been deposited within the protein data bank with accession codes: 5NA5, 5NAB, 5NAE, 5NAG, 5NAH, 5NAK.

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

## Acknowledgements

We thank Michelle Pemberton for assistance with the Rapidfire data generation. D.J.M. is
supported by a Clinician Scientist Fellowship from the Health Foundation/Academy of
Medical Sciences, the Medical Research Council Developmental Pathway Funding
Stream and Wellcome Trust Institutional Strategic Support Fund.

## Author contributions

All authors developed and refined the experimental hypothesis and experimental design.
E.M.C, C.I.H., P.H. expressed and purified proteins for biochemical experiments
designed and performed by J.P.H. and C.H. A.L.W. and J.L. devised the medicinal
chemistry. P.R., M.R.D.T., C.G.M. and C.C. designed, conducted and analysed the
crystallography, respectively. D.J.M., D.S.H., S.P.W., I.U. and J.L. led the University of
Edinburgh/GlaxoSmithKline Discovery Partnership with Academia collaboration. All
authors contributed to other aspects of data analysis and/or interpretation. C.C. and
J.P.H wrote the manuscript, with contributions and revisions by all coauthors.

## Additional information

naturecommunications

**Competing interests:** GlaxoSmithKline holds patents on the molecules described in the
paper. J.P.H., P.R., E.M.C., C.H., C.I.H., D.S.H., P.H., J.L., I.U., A.L.W. and C.C. are
employees of GlaxoSmithKline but do not receive any direct remuneration based on
the success of this specific project. M.R.D.T., D.J.M., S.P.W. and C.G.M. work for the
University of Edinburgh which is engaged in a Discovery Partnership with Academia
collaboration with GlaxoSmithKline by which the University of Edinburgh receives
milestone according to the phase of development of the KMO inhibitors described and
would receive royalty payments on any resulting commercial sales. These payments are
governed by the University of Edinburgh revenue sharing policy.

**Reprints and permission** information is available online at http://npg.nature.com/
reprintsandpermissions/

differentiated inhibitors of the acute pancreatitis target kynurenine-3-monooxygenase.
*Nat. Commun.* **8**, 15827 doi: 10.1038/ncomms15827 (2017).

**Publisher's note:** Springer Nature remains neutral with regard to jurisdictional claims in
published maps and institutional affiliations.

