## [Peer Review File · Nature Communications]

Reviewer #1 (Remarks to the Author):

Kynurenine monoxygenase (KMO) is an emerging drug target enzyme for a number of serious diseases. This paper describes exciting new high resolution crystal structures of KMO complexed with substrate and high potency inhibitors. These results will be useful in the future development of novel inhibitors which may become clinical candidates. The paper is clear and concise. The discovery of the flipped FAD conformation as a high affinity form may have broad implications for design of flavoenzyme inhibitors. The experimental portion describes the work sufficiently. I have only a few comments and minor corrections.

1. In the structure shown in Figure 2, how does the C-terminal helical domain compare in sequence to the C-terminal membrane anchor domain of the mammalian enzyme? Is it more hydrophilic to make it soluble? Does this structure explain why efforts to improve solubility of pig KMO by truncation resulted in inactive enzyme?
2. In Figure 2 legend in the text (but not at the end of the manuscript) the K_m values are given in mM instead of μ M.
3. The biological names of the organisms, *Pseudomonas fluorescens* and *Saccharomyces cerevisiae*, should be in italics throughout.
4. Line 117, references 16,16 should probably be 15,16.

Reviewer #2 (Remarks to the Author):

The GSK and University of Edinburgh teams have already published the ability of their lead compound (GSK180) to prevent organ failure in acute pancreatitis (Nat Med Feb 2016).

Inhibiting KMO has become a really hot topic for several drug companies.

This manuscript is very well written.

The scientific and chemistry approaches are excellent.

I have read this manuscript twice and couldn't find any issue.

Very professional and clearly a lot of work and editing behind.

Reviewer #3 (Remarks to the Author):

Hutchinson et al. 17.02 .2017

Hutchinson et al. presented the structure of kynurenine-3-monoxygenase (KMO) in complex with the substrate L-kynurenine and two types of inhibitors complemented with comprehensive kinetic studies. KMO is involved in neurodegenerative diseases and acute pancreatitis and therefore of pharmaceutical interest.

The manuscript is, in principle, relevant for publication in Nat. Comm. because (I) some of the inhibitor analysed for KMO are already in the stage for preclinical evaluation, (II) the structural in combination with kinetic data convincingly rationalize on an atomic level the unusual properties of the type II inhibitors (pm affinity, increased residence time, no hydrogen peroxide), (III) the tilting of the isoalloxazine ring represents a special unprecedented feature and (IV) the data stimulates the development of novel inhibitors for KMO and other flavoenzymes.

An interesting aspect of the work, in addition, is that the experimental finding of an empty water-filled pocket adjacent to the type I inhibitor appeared to initiate the synthesis of the beneficial type II inhibitors which do, however, not bind in the expected manner. The presented data also suggest that the tilting of the isoalloxazine ring (the key for the success of type II inhibitors) is

energetically feasible because a similar conformation exists upon NADPH binding during the energetically optimized catalytic process. Fortunately, the isoalloxazine – pyridyl π -stack interactions do not only increase the affinity of the inhibitor but also prevents NADPH binding and thus hydrogen peroxide formation. This observation also implicates that either the type II inhibitor or both NADPH and L-kyn can be bound. These aspects are relevant for drug designer and it would be interesting whether the binding position of type II inhibitors can be also obtained by up-to-date calculations.

The presented structural and kinetic data appear to be methodically sound. They also convincingly support the conclusion of the authors that the structural analysis of the complete three-domain KMO is crucial for understanding substrate and inhibitor binding and that the *Pseudomonas fluorescens* KMO is a suitable surrogate for human KMO.

However, the manuscript is currently not written in a suitable quality for publication. It can be considerably condensed without losing quality. Some experiments are described in the text and the legend in great detail and some aspects (i.e. line 108-117 and line 275-286) can be significantly shortened or even omitted. The discussion is essentially a summary and the discussion is already included in the results (i.e. modelling of NADH binding) which is not incorrect but should be indicated by a "Results and Discussion" paragraph (or alternatively consequently separated).

Minor points

Fig. 1, a "+" at the carbon bound to amine or $C=NH_2^+$ double bond is missing.

Line 58, "symptoms resolve within a week or so" to "symptoms mostly disappear within on week"

Line 61, "so this represents" to "which indicates"

Line 79, "*Pseudomonas fluorescens*" to "*Pseudomonas fluorescens*" and later "*Saccharomyces*"

Line 93, "hydrophobic small side chains" to "small hydrophobic side chains"

Fig. 2, a is not essential

Line 226, "demonstrates ... new inhibitors" please rephrase

Fig.4, b, d and f are not essential

Fig. 6, c, d supplementary data (similar as Fig. 5 d, e)

Line 293, "which suggests that the inhibitor and NADPH bind to different states of the enzyme"

The meaning of "states" is not clear enough. Do you mean sites? Perhaps, the concentrations of L-kyn and NADPH have to be concertedly increased to compete with the type II inhibitor.

Reply to Referees' comments on Nature communications manuscript NCOMMS-17-00514

Reviewer #1 (Remarks to the Author):

Kynurenine monooxygenase (KMO) is an emerging drug target enzyme for a number of serious diseases. This paper describes exciting new high resolution crystal structures of KMO complexed with substrate and high potency inhibitors. These results will be useful in the future development of novel inhibitors which may become clinical candidates. The paper is clear and concise. The discovery of the flipped FAD conformation as a high affinity form may have broad implications for design of flavoenzyme inhibitors. The experimental portion describes the work sufficiently. I have only a few comments and minor corrections.

1. In the structure shown in Figure 2, how does the C-terminal helical domain compare in sequence to the C-terminal membrane anchor domain of the mammalian enzyme? Is it more hydrophilic to make it soluble? Does this structure explain why efforts to improve solubility of pig KMO by truncation resulted in inactive enzyme?
2. In Figure 2 legend in the text (but not at the end of the manuscript) the K_m values are given in mM instead of μM .
3. The biological names of the organisms, *Pseudomonas fluorescens* and *Saccharomyces cerevisiae*, should be in italics throughout.
4. Line 117, references 16,16 should probably be 15,16.

Author's reply:

We thank the reviewer for their excitement and agree that these new insights and structures will prove of significant value in advancing this area. In specific response to their minor comments and corrections:

1. The predicted transmembrane region of human KMO (425-446) contains no strongly hydrophilic residues, and an even distribution of hydrophobic residues when viewed in a helical representation, consistent with the predicted role. The equivalent region in Pf KMO (431-452) contains significant numbers of highly hydrophilic residues and fewer hydrophobic residues. Additionally, pairs of basic side chains present on either side of the human transmembrane region, which are proposed to facilitate membrane targeting, are absent in Pf KMO. The relationship between the activity of truncates and mutations in the far C-terminal region of mammalian KMO proteins appears complex resulting in differences in KMO localisation and/or activity. However, as our structure now shows domain 3 is important in completing the environment of the active site, we hypothesise efforts to improve solubility of pig KMO by truncation may have inadvertently resulted in inactive enzyme as the C-term region within mammalian KMO stabilises the structure or orientation of domain 3 relative to the active site.
2. We have ensured that the Figure 2 legend has the K_M values corrected as μM in all instances.
3. We have italicised *Pseudomonas fluorescens* and *Saccharomyces cerevisiae* throughout.
4. Reference in 117 has been corrected from 16,16 to 15,16.

Reviewer #2 (Remarks to the Author):

The GSK and University of Edinburgh teams have already published the ability of their lead compound (GSK180) to prevent organ failure in acute pancreatitis (Nat Med Feb 2016).

Inhibiting KMO has become a really hot topic for several drug companies.

This manuscript is very well written.

The scientific and chemistry approaches are excellent.

I have read this manuscript twice and couldn't find any issue.

Very professional and clearly a lot of work and editing behind.

Author's reply:

We are delighted that the reviewer's complementary comments and consciousness in reading the manuscript twice. We hope he enjoyed this enough to read a third time when it comes out within the journal.

Reviewer #3 (Remarks to the Author):

Hutchinson et al. presented the structure of kynurenine-3-monooxygenase (KMO) in complex with the substrate L-kynurenine and two types of inhibitors complemented with comprehensive kinetic studies. KMO is involved in neurodegenerative diseases and acute pancreatitis and therefore of pharmaceutical interest.

The manuscript is, in principle, relevant for publication in Nat. Comm. because (I) some of the inhibitor analysed for KMO are already in the stage for preclinical evaluation, (II) the structural in combination with kinetic data convincingly rationalize on an atomic level the unusual properties of the type II inhibitors (pm affinity, increased residence time, no hydrogen peroxide), (III) the tilting of the isoalloxazine ring represents a special unprecedented feature and (IV) the data stimulates the development of novel inhibitors for KMO and other flavoenzymes.

An interesting aspect of the work, in addition, is that the experimental finding of an empty water-filled pocket adjacent to the type I inhibitor appeared to initiate the synthesis of the beneficial type II inhibitors which do, however, not bind in the expected manner. The presented data also suggest that the tilting of the isoalloxazine ring (the key for the success of type II inhibitors) is energetically feasible because a similar conformation exists upon NADPH binding during the energetically optimized catalytic process. Fortunately, the isoalloxazine – pyridyl π -stack interactions do not only increase the affinity of the inhibitor but also prevents NADPH binding and thus hydrogen peroxide formation. This observation also implicates that either the type II inhibitor or both NADPH and L-kyn can be bound. These aspects are relevant for drug designer and it would be interesting whether the binding position of type II inhibitors can be also obtained by up-to-date calculations.

The presented structural and kinetic data appear to be methodically sound. They also convincingly support the conclusion of the authors that the structural analysis of the complete three-domain KMO is crucial for understanding substrate and inhibitor binding and that the *Pseudomonas fluorescens* KMO is a suitable surrogate for human KMO.

However, the manuscript is currently not written in a suitable quality for publication. It can be considerable condensed without losing quality. Some experiments are described in the text and the legend in great detail and some aspects (i.e. line 108-117 and line 275-286) can be significantly shortened or even omitted. The discussion is essentially a summary and the discussion is already included in the results (i.e. modelling of NADH binding) which is not incorrect but should be indicated by a "Results and Discussion" paragraph (or alternatively consequently separated).

Minor points

Fig. 1, a "+" at the carbon bound to amine or C=NH₂⁺ double bond is missing.

Line 58, "symptoms resolve within a week or so" to "symptoms mostly disappear within on week"

Line 61, "so this represents" to "which indicates"

Line 79, "*Pseudomonas fluorescens*" to "*Pseudomonas fluorescens*" and later "*Saccharomyces*"

Line 93, "hydrophobic small side chains" to "small hydrophobic side chains"

Fig. 2, a is not essential

Line 226, "demonstrates ... new inhibitors" please rephrase

Fig.4, b, d and f are not essential

Fig. 6, c, d supplementary data (similar as Fig. 5 d, e)

Line 293, "which suggests that the inhibitor and NADPH bind to different states of the enzyme"

The meaning of "states" is not clear enough. Do you mean sites? Perhaps, the concentrations of L-kyn and NADPH have to be concertedly increased to compete with the type II inhibitor.

Author's reply:

We thank this reviewer for his careful reading and have addressed many of the minor points fully as documented below.

1. Fig. 1, a "+" at the carbon bound to amine or C=NH₂⁺ double bond is missing.
This has been corrected.
2. Line 58, "symptoms resolve within a week or so" to "symptoms mostly disappear within on week"
We feel the phrase used more accurately reflect the resolution of symptoms so prefer not to alter this wording.
3. Line 61, "so this represents" to "which indicates"
This has been changed as the reviewer suggested.
4. Line 79, "Pseudomonas fluorescens" to "*Pseudomonas fluorescens*" and later "Saccharomyces"
We have italicised *Pseudomonas fluorescens* and *Saccharomyces cerevisiae* throughout.
5. Line 93, "hydrophobic small side chains" to "small hydrophobic side chains"
This has been changed as the reviewer suggested.
6. Fig. 2, a is not essential
Whilst we agree Fig2a is not absolutly essential, we believe readers will find this depiction useful to navigate between sequence and domain structure, so we would like to retain this very small figure.
7. Line 226, "demonstrates ... new inhibitors" please rephrase
We have substituted the word demonstrates by illustrates.
8. Fig.4, b, d and f are not essential
We feel this series of figures help illustrate the narrative so would like to retain these figures.
9. Fig. 6, c, d supplementary data (similar as Fig. 5 d, e)
We have retained these subfigures in the supplementary as it is useful for readers to see the full set of underlying data and alongside the derived summary plots within a single page.
10. Line 293, "which suggests that the inhibitor and NADPH bind to different states of the enzyme"
The meaning of "states" is not clear enough. Do you mean sites? Perhaps, the concentrations of L-kyn and NADPH have to be concertedly increased to compete with the type II inhibitor.
We have deliberately used the term state to NOT mean site. Unfortunately as the human enzyme cannot be purified to homogeneity, but is used from enriched lysate, many of the elegant and complex kinetic experiments possible with purified FMOs are not accessible to the human enzyme.

Finally, we value and welcome the reviewer's many helpful suggestions, however, we respectfully disagree with the comment that the submitted manuscript was *not written in a suitable quality for publication*, especially given the contrary complementary remarks of the other reviewers on the same points that this reviewer raises (e.g. clarity and conciseness of the writing). We feel this may be a difference in style, so were pleased that despite this stylistic difference the reviewer found the science within the paper sufficiently important and interesting for Nature communs. and our conclusions "sound".